# Identification of Filamentous Fungi by MALDI-TOF Mass Spectrometry: Evaluation of Three Different Sample Preparation Methods and Validation of an In-House Species Cutoff

**DOI:** 10.3390/jof8040383

**Published:** 2022-04-10

**Authors:** Claudia Honsig, Brigitte Selitsch, Marlene Hollenstein, Matthias G. Vossen, Kathrin Spettel, Birgit Willinger

**Affiliations:** 1Division of Clinical Microbiology, Department of Laboratory Medicine, Medical University of Vienna, 1090 Vienna, Austria; claudia.honsig@meduniwien.ac.at (C.H.); brigitte.selitsch@meduniwien.ac.at (B.S.); kathrin.spettel@meduniwien.ac.at (K.S.); 2Department of Laboratory Medicine, Medical University of Vienna, 1090 Vienna, Austria; marlene.hollenstein@meduniwien.ac.at; 3Division of Infectious Diseases and Tropical Medicine, Department of Medicine I, Medical University of Vienna, 1090 Vienna, Austria; matthias.vossen@meduniwien.ac.at

**Keywords:** filamentous fungi, identification, MALDI-TOF MS

## Abstract

Invasive infections caused by filamentous fungi constitute a leading cause of morbidity and mortality in immunocompromised patients. Rapid and reliable identification of filamentous fungi is essential for the early initiation of appropriate treatment. In the present study, 230 filamentous fungi isolates identified by conventional methods were investigated using MALDI-TOF MS (Bruker Daltonics, Bremen, Germany) in combination with the Filamentous Fungi Library 3.0 provided by the manufacturer. Three different sample preparation methods were applied as recommended by the manufacturer and identification rates were compared using the criteria provided by the manufacturer. Application of the more time-consuming sample preparation methods clearly improved identification at the species level. Thus, the identification rate increased from 48.9% using the simplest method to 76.1% with the most laborious procedure. Misidentifications did not occur. Furthermore, the reliability of an in-house threshold for species identification was investigated. The reduced threshold increased the rate of isolates correctly identified at the species level by up to 86.4%. As no misidentification was made at the genus level and only one misidentification of minor significance occurred at the species level, this threshold could be validated for routine use in our laboratory. In conclusion, regarding the high identification rates achieved, this commercial platform proved suitable for implementation in routine diagnosis.

## 1. Introduction

Invasive infections caused by filamentous fungi constitute a leading cause of morbidity and mortality among the rising number of immunocompromised patients. Patients with haematologic malignancies, haematopoietic stem cell transplant recipients, solid organ transplant recipients and critically ill patients are especially at risk. Rapid and reliable identification of clinical isolates is highly important for the early initiation of effective antifungal treatment [1].

Traditionally, identification of filamentous fungi is based on macroscopic and microscopic features of isolates and requires a skilled medical mycologist [2]. In addition, molecular techniques such as DNA amplification by polymerase chain reaction (PCR) and DNA sequencing are used to identify clinical isolates. As these methods are quite time-consuming and expensive, their use in routine diagnostics is limited [3]. Matrix-assisted laser desorption/ionization time-of-flight (MALDI-TOF) mass spectrometry (MS) is a widely used technique for rapid and accurate identification of bacteria and yeasts and is gradually replacing conventional identification methods [4,5,6]. Recently, this technique has emerged as an additional tool to identify isolates of filamentous fungi [6,7,8]. The benefits of the implementation of extensive in-house libraries for this purpose have been reported before in several publications [6,9,10,11,12]. In the present study, 230 filamentous fungi isolates were investigated using MALDI-TOF MS (Bruker Daltonics, Bremen, Germany) in combination with the Filamentous Fungi Library 3.0 provided by the manufacturer. According to the manufacturer, three different sample preparation methods can be applied—the basic and fastest sample preparation procedures are the direct transfer and extended direct transfer procedures and these should be performed first. The extraction is the second option which increases the identification rate, and liquid cultivation is described as the procedure with the highest identification rate.

The aim of the present study was to evaluate the impact of these three different methods of sample preparation on the rate of identification of filamentous fungi isolates at the genus and species levels, respectively, and to assess their suitability for incorporation into the routine diagnosis workflow. To our knowledge, a comparison of the different sample preparation methods performed according to the manufacturer’s instructions using a commercially available filamentous fungi library has not been carried out so far.

## 2. Materials and Methods

### 2.1. Conventional Methods Used for the Identification of Fungal Isolates

In the present study, a collection of 230 filamentous fungi isolates consisting of 185 clinical isolates and 45 reference strains, respectively, was investigated retrospectively (Table 1). Dermatophytes were not included in this collection. The clinical isolates had been identified based on their micro- and macromorphological features. Morphological identification at the species level had been achieved for 206 isolates, while 24 isolates had been identified only at the genus level. For 18 isolates, sequence analysis of the ITS2 region had been performed in order to verify the result of the phenotypic identification.

### 2.2. Sequence Analysis of the ITS2 Region

After subculturing the fungal isolates, DNA was extracted using thermal lysis. Panfungal real-time PCR and sequence analysis of the ITS2 region was performed as described previously [13].

### 2.3. MALDI-TOF Mass Spectrometry

The fungal strains were cultured on Sabouraud dextrose agar with chloramphenicol and gentamicin (Becton Dickinson GmbH, Heidelberg, Germany) and incubated at 30 °C for up to one week, until sufficient growth was observed. The isolates were subjected to MALDI-TOF MS analysis using a Bruker MALDI Biotyper microflex LT (Bruker Daltonics, Bremen, Germany).

### 2.4. Sample Preparation

For each of the isolates, the following sample preparation procedures for MALDI-TOF MS were performed as described by the manufacturer: (i) extended direct transfer, (ii) extraction and (iii) liquid cultivation sample preparation (Standard Operating Procedure Cultivation and Sample Preparation for Filamentous Fungi, Revision E, March 2019, Bruker Daltonics, Bremen, Germany).

(i) **Extended direct transfer procedure**: In brief, for the basic sample preparation procedure, the extended direct transfer, front mycelium was harvested from Sabouraud plates, smeared on the target plate and overlaid with 1 µL 70% formic acid (which is the only difference to the direct extraction procedure), and in a second step overlaid with 1 µL matrix solution. The matrix-overlaid sample was air-dried and then inserted into the MALDI-TOF mass spectrometer.

(ii) **Extraction sample preparation procedure**: The extraction was performed to increase the identification rate due to better protein accessibility. Briefly, the front mycelium was transferred into 1 mL HPLC water, pelleted by centrifugation and the pellet was again dissolved in 300 µL HPLC water. Then, 900 µL ethanol was added and after centrifugation the ethanol was removed and the sample was air-dried. Depending on the size, the pellet was resuspended in 10–100 µL 70% formic acid. The same amount of acetonitrile was then added and centrifuged again. Afterwards, 1 µL of the supernatant was pipetted onto the MALDI target, overlaid with 1 µL of HCCA matrix and analyzed with MALDI-TOF.

(iii) **Liquid cultivation sample preparation procedure**: In this procedure, fungi were grown in liquid Sabouraud medium before extraction using the rotator, overnight or until sufficient growth was observed. Then, 1.5 mL from the sedimented liquid culture was prepared for the MALDI-TOF MS measurements following the extraction sample preparation procedure described above.

Each of the three different preparations was placed in two sample positions on a MALDI Biotarget plate.

### 2.5. Interpretation of Results

The resulting spectra were assessed using the Bruker Filamentous Fungi Library 3.0., which contains 180 species and an additional 10 genera without species identification. In total, 62 genera are included in the database. The logscore values were interpreted according to the manufacturer: identification scores ≥2.0 indicated high confidence identification, scores of ≥1.70–1.99 indicated low confidence identification and scores <1.70 indicated no reliable identification. Thus, for each of the preparation methods, identification of the isolates was accepted if at least one of the duplicate spots yielded a score ≥1.70. If the two score values yielded for each spot differed, the higher one was considered as the final result. The identification by MALDI-TOF MS was considered correct when it was concordant with the phenotypic identification.

In order to assess whether lowering the species cutoff may increase the identification rate without introducing misidentifications, a threshold of ≥1.70 was applied for species identification, as proposed previously [6,14]. The results obtained using this “in-house” cutoff were compared with the results for the phenotypic characterization of the isolates. The species was considered as correctly identified whenever one of the scores was ≥1.70. When there was more than one identification with a score ≥1.70 all of the results had to yield the same species name before it was considered that a correct identification had been made.

## 3. Results

### 3.1. Isolates Identified at the Species Level by Conventional Methods

Out of 206 isolates being identified at the species level, 184 isolates were included in the Filamentous Fungi Library 3.0. These isolates comprised 35 different species belonging to 14 different genera (Table 1).

As expected, and as stated by the manufacturer, overall, extraction increased the identification rate compared with extended direct transfer, and the most efficient identification was achieved with the most laborious procedure, namely, liquid cultivation.

(i) Using the extended direct transfer method, 135 of the 184 isolates included (73.4%) could be identified. Identification at species level was achieved for 90 isolates (48.9%) and identification at the genus level for 45 (24.5%) isolates.

(ii) When applying the extraction method, 157 isolates (85.3%) could be identified: 114 isolates (62.0%) at the species level and 43 (23.4%) at the genus level.

(iii) Using the liquid cultivation method, identification of 161 isolates (87.5%) was achieved: 140 isolates (76.1%) could be identified at the species level and 21 (11.4%) only at the genus level.

The results according to the thresholds provided by the manufacturer are presented in detail in Table 2. All of the isolates that were identified at the species level or at the genus level, respectively, were correctly identified. None of the isolates not included in the library was misidentified. For these isolates, no identification was obtained.

When applying the threshold of 1.70 for species identification, a remarkable increase in identifications at the species level was observed (Table 3). Almost all of the isolates identified at the genus level only when applying the manufacturer’s recommendations could be identified at the species level without any misidentifications.

When using the extended direct transfer procedure, identification at species level rose from 90 isolates (48.9%) according to the Bruker classification of results to 134 isolates (72.8%) with the in-house threshold. Only one isolate of *Aspergillus section Usti* remained identified at the genus level only.

The extraction procedure achieved identification at the species level in 114 (62.0%) according to Bruker versus 155 (84.2%) isolates when applying the in-house threshold. Out of five isolates of *Penicillium chrysogenum,* only one strain could not be identified at the species level.

Liquid cultivation reached the highest rates of species identification: 140 (76.1%) according to Bruker versus 159 isolates (86.4%) with our in-house threshold. Two isolates, i.e., one out of three *Aspergillus candidus* isolates and one out of eight strains of *Fusarium oxysporum complex*, respectively, did not fulfill the in-house criteria for identification at species level.

All of the identifications at the species level were consistent with those obtained with conventional methods and sequence analysis, respectively. As no misidentifications occurred, the analyses carried out were rated as sufficient to validate this threshold for routine use in our laboratory.

### 3.2. Isolates Not Included in the Filamentous Fungi Library 3.0

Twenty-two isolates were not included in the Filamentous Fungi Library 3.0 as they belonged to the following six genera: *Acremonium, Alternaria, Bipolaris, Cunninghamella, Rhizomucor* and *Scytalidium*. In 21 cases, no identification was achieved with any of the three extraction methods. Only in one case could a result be achieved by MALDI-TOF MS. The respective strain identified as *Rhizomucor miehei* was correctly identified at genus level with the liquid cultivation method when applying the manufacturer’s criteria for interpretation of the results. When using the in-house criteria, this isolate was identified at the species level as *Rhizomucor pusillus*, which is the only *Rhizomucor* species included in the database.

### 3.3. Isolates Identified at the Genus Level by Conventional Methods

Twenty-four strains identified only at the genus level belonged to *Alternaria*, *Aspergillus*, *Bipolaris*, *Cladosporium*, *Cunninghamella*, *Fusarium*, *Lichtheimia*, *Mucor*, *Penicillium*, *Phaeoacremonium* and *Trichoderma* (Table 1).

The genus *Bipolaris* was not included in the database and therefore, as expected, the *Bipolaris species* from our collection could not be identified by MALDI-TOF-MS. However, no false identification was achieved.

The remaining 23 isolates were included in the database. When applying the manufacturer’s criteria, 11 of these isolates were identified at the species level and 4 of the isolates (*Lichtheimia* sp. *n* = 1 and *Trichoderma* sp. *n* = 3) only at the genus level. However, as it was not possible to identify the species either by conventional methods or sequence analysis, only correctly identified genera were confirmed. Eight isolates belonging to the genera *Cladosporium* (*n* = 2), *Cunninghamella* (*n* = 1), *Penicillium* (*n* = 2) and *Trichoderma* (*n* = 3) could not be identified using any of the three sample preparation methods. The results of identification by MALDI-TOF MS following the three different sample preparation procedures and using the manufacturer’s interpretation criteria are shown in detail in Table 4. As shown in Table 5, when applying the threshold of 1.70 for species identification, a remarkable increase in identifications at the species level was observed. Overall, for the isolates identified at the genus level by conventional methods, no misidentification occurred.

## 4. Discussion

Despite being slow and lacking sensitivity, the isolation of a fungal pathogen in culture remains the gold standard for diagnosing fungal disease and is essential for antifungal therapeutic management. As antifungal susceptibility patterns can vary substantially between species, a rapid and reliable identification of filamentous fungi is essential for appropriate patient management and improving patient outcomes [15]. Until recently, the identification of filamentous fungi had been based on macroscopic and microscopic characteristics. This is a slow and labour-intensive procedure and relies on the knowledge of highly trained experts [15]. To allow a more rapid and accurate identification of fungi, MALDI-TOF MS has been applied for the identification of yeast and moulds isolated from clinical specimens, involving minimal sample preparation and with results available in minutes [16].

Filamentous fungi identification rates with MALDI-TOF MS vary depending on the library used, the variety of isolates tested and the sample preparation procedure. In previous studies, it has already been shown that MALDI-TOF MS is suitable for use in routine laboratory settings if sample preparation procedures are optimized and appropriate databases are available [6,11,17]. Commercial databases have been shown to require optimization and so far have mainly been used in combination with in-house libraries [14]. As reported by Stein et al. [17], reliable identification of filamentous fungi by MALDI-TOF MS has become a reality in a few laboratories only recently and is not yet a widespread routine method.

In the present study, we used the commercial Bruker platform together with the Bruker MBT Filamentous Fungi Library 3.0 and standardized sample preparation procedures as recommended by the manufacturer for the investigation of 230 filamentous fungi isolates. The majority of isolates were covered by the database, and those that were not included in the database served as controls in order to rule out possible misidentifications. Three different sample preparation procedures for MALDI-TOF MS were compared in order to assess the impact on the identification rates and the suitability for routine use. The simplest protocol, extended direct transfer, is a rapid procedure which takes around 5 min. If this method fails, an extraction procedure is proposed, which requires an additional 10 min. As a final alternative when the extraction procedure does not result in species identification, culture in a liquid medium should be performed. This sample preparation procedure extends the identification time by at least 24 h.

Our data demonstrate that it is useful to follow the sequence of procedures suggested by Bruker, as application of the more time-consuming procedures clearly increases identification rates at the species level. Of the genera *Alternaria*, *Exophilia*, *Fusarium*, *Paecilomyces* and *Purpureocillium*, all the isolates included in the study were correctly identified. The sample size, however, was very small for some of these genera. *Aspergillus*, the genus with by far the most isolates included in this study, turned out to be excellently identifiable. In addition, good identification rates were also achieved for *Mucorales*, such as *Lichtheimia*, *Mucor*, *Rhizomucor* and *Rhizopus*. Misidentifications did not occur, and only rarely did the identification of isolates included in the library fail. This was only seen with the genera *Cladosporium* and *Cunninghamella*. *Penicillium* and *Schizophyllum* species, as expected, turned out to be poorly identifiable at the species level.

Despite the limited number of isolates investigated in our study, our data clearly show that correct species identification of clinically important moulds can be achieved at a high level using this commercially available platform and database.

Overall, the remarkable increase in identifications achieved with the liquid cultivation procedure, especially at the species level but also at the genus level, justifies the use of this method in case the other procedures fail.

Application of 1.70 as a threshold for species identification clearly increased the number of isolates correctly identified at the species level and resulted in only one misidentification, which could be explained by the close relationship of the species in question, *Rhizomucor pusillus* and *Rhizomucor miehei* [18]. Therefore, this misidentification was not considered a serious error in the database. Based on these excellent results, we were able to validate the 1.70 cutoff as an in-house threshold for species identification.

In conclusion, our data suggest that the Bruker Biotyper in conjunction with the commercially available Filamentous Fungi Library 3.0 and the interpretation criteria proposed by Bruker is suited for implementation in routine diagnosis. Identification rates when following the manufacturer’s recommendations have proven useful in the routine workflow and lowering the species cutoff has been shown to achieve considerably higher rates of isolates correctly identified at the species level. No misidentification was observed at the genus level and only one misidentification of minor significance occurred at the species level. Based on these results, the lower threshold of 1.70 for identification of filamentous fungi at the species level could be implemented in the routine diagnostic process in our laboratory. However, optimization of the library by increasing the number of included reference spectra would be beneficial. The manufacturer has only recently released a new version of the database, the Filamentous Fungi Library V.4.0 as well as the Filamentous Fungi Module to improve the identification of filamentous fungi. Further studies will be needed to assess the impact of the changes and extensions associated with this commercial filamentous fungi library.

## Figures and Tables

**Table 1 jof-08-00383-t001:** Isolates included in the study (*n* = 230). Clinical isolates (*n* = 185) identified with high confidence (at species level) (*n* = 161) and only identified with low confidence (at genus level) (*n* = 24), respectively. Reference strains (*n* = 45). Abbreviations: n: number; library: Bruker Filamentous Fungi Library 3.0; ✓: included in the library; -: not included in the library.

Genus	*n*	Species	Included in Library	n Clinical Isolates	n Reference Strains
*Acremonium*	6	*A. blochii*	-	1	
		*A. kiliense*	-	1	
		*A. polychromum*	-	1	
		*A. strictum*	-	1	2
*Alternaria*	6	*A. alternata*	✓	3	
		*A. infectoria*	-	1	
		*A. species*	✓	2	
*A* *spergillus*	81	*A. candidus complex*	✓	1	2
		*A. fumigatus*	✓	15	
		*A. lentulus*	✓	1	
		*A. nidulans*	✓	9	
		*A. niger*	✓	15	
		*A. oryzae*	✓	16	
		*A. sclerotiorum*	✓	1	
		*A. species*	✓	1	
		*A. terreus*	✓	13	
		*A. ustus*	✓	5	
		*A. versicolor*	✓		2
*Bipolaris*	5	*B. australiensis*	-	1	
		*B. hawaiiensis*	-		3
		*B. species*	-	1	
*Cladosporium*	6	*C. cladosporioides*	✓		1
		*C. herbarum*	✓	1	1
		*C. sphaerospermum*	✓		1
		*C. species*	✓	2	
*Cunninghamella*	8	*C. bertholletiae*	-		6
		*C. elegans*	✓	1	
		*C. species*	✓	1	
*Exophilia*	5	*E. dermatitidis*	✓	5	
*Fusarium*	24	*F. dimerum*	✓	3	
		*F. oxysporum complex*	✓	8	
		*F. solani*	✓	8	
		*F. species*	✓	5	
*Lichtheimia*	14	*L. corymbifera*	✓	11	2
		*L. species*	✓	1	
*Mucor*	9	*M. circinelloides*	✓	5	2
		*M. hiemalis*	✓		1
		*M. species*	✓	1	
*Paecilomyces*	3	*P. variotii*	✓	2	1
*Penicillium*	13	*P. chrysogenum*	✓	3	2
		*P. citreonigrum*	✓	1	
		*P. citrinum*	✓	1	
		*P. glabrum*	✓	1	
		*P. purpurogenum*	✓	1	1
		*P. species*	✓	3	
*Phaeoacremonium*	1	*P*. *species*	-	1	
*Purpureocillium*	8	*P. lilacinum*	✓	7	1
*Rhizomucor*	5	*R. miehei*	-	1	1
		*R. pusillus*	✓	2	1
*Rhizopus*	11	*R. microsporus*	✓	2	2
		*R. oryzae*	✓	5	2
*Scedosporium*	12	*S. apiospermum complex*	✓		6
		*S. prolificans*	-	2	4
*Schizophyllum*	4	*S. commune*	✓	4	
*Scytalidium*	3	*S. dimiduatum*	-	1	1
		*S. hyalinum*	-	1	
*Trichoderma*	6	*T. species*	✓	6	
total	230			185	45

**Table 2 jof-08-00383-t002:** Clinical isolates identified at the species level and reference strains included in the Filamentous Fungi Library 3.0 (*n* = 184). Results of the MALDI-TOF MS using three different sample preparation procedures. Interpretation of results, i.e., identification with high confidence (at the species level) or low confidence (at the genus level), according to the manufacturer’s recommendations. Abbreviations: n: number, ID hc.: number of isolates identified with high confidence (at the species level), ID lc.: number of isolates identified with low confidence (at the genus level), no ID: number of isolates not identified.

Clinical Isolates and Reference Strains				MALDI-TOF MS								
				Extended Direct Transfer			Extraction			Liquid Cultivation		
Genus	*n*	Species	*n*	ID hc.	ID lc.	no ID	ID hc.	ID lc.	no ID	ID hc.	ID lc.	no ID
*Alternaria*	3	*A. alternata*	3	2	1	-	2	1	-	2	1	-
*Aspergillus*	80	*A. candidus complex*	3	1	1	1	-	1	2	2	1	-
		*A. fumigatus*	15	7	5	3	12	1	2	15	-	-
		*A. lentulus*	1	-	1	-	1	-	-	-	1	-
		*A. nidulans*	9	5	1	3	6	2	1	9	-	-
		*A. niger*	15	10	4	1	14	-	1	15	-	-
		*A. oryzae*	16	9	4	3	6	10	-	13	3	-
		*A. sclerotiorum*	1	1	-	-	1	-	-	1	-	-
		*A. terreus*	13	9	2	2	9	2	2	8	1	4
		*A. ustus*	5	4	1	-	5	-	-	5	-	-
		*A. versicolor*	2	2	-	-	-	1	1	2	-	-
*Cladosporium*	4	*C. cladosporioides*	1	-	-	1	-	-	1	-	-	1
		*C. herbarum*	2	-	-	2	-	-	2	-	-	2
		*C. sphaerospermum*	1	-	-	1	-	-	1	-	-	1
*Exophilia*	5	*E. dermatitidis*	5	5	-	-	3	1	1	5	-	-
*Fusarium*	19	*F. dimerum*	3	-	-	3	-	2	1	3	-	-
		*F. oxysporum complex*	8	1	6	1	4	2	2	7	1	-
		*F. solani*	8	4	3	1	4	4	-	8	-	-
*Paecilomyces*	3	*P. variotii*	3	3	-	-	3	-	-	3	-	-
*Penicillium*	10	*P. chrysogenum*	5	-	1	4	-	3	2	2	2	1
		*P. citreoni*	1	-	-	1	-	-	1	-	-	1
		*P. citrinum*	1	-	1	-	-	-	1	1	-	-
		*P. glabrum*	1	-	1	-	-	1	-	-	1	-
		*P. purpurogenum*	2	-	-	2	1	-	1	1	-	1
*Purpureocillium*	8	*P. lilacinum*	8	5	2	1	5	3	-	8	-	-
*Scedosporium*	12	*S. apiospermum complex*	12	9	1	2	10	-	2	9	-	3
*Schizophyllum*	4	*S. commune*	4	-	-	4	1	2	1	-	3	1
*Mucorales*												
*Cunninghamella*	1	*C. elegans*	1	-	-	1	-	-	1	-	-	1
*Lichtheimia*	13	*L. corymbifera*	13	2	6	5	9	3	1	7	2	4
*Mucor*	8	*M. circinelloides*	7	5	1	1	6	1	-	5	-	2
		*M. hiemalis*	1	-	-	1	-	1	-	-	1	-
*Rhizomucor*	3	*R. pusillus*	3	3	-	-	3	-	-	3	-	-
*Rhizopus*	11	*R. arrhizus*	7	2	2	3	6	1	-	3	4	-
		*R. microsporus*	4	1	1	2	3	1	-	3	-	1
Total n			184	90	45	49	114	43	27	140	21	23
Total %			100%	48.9%	24.5%	26.6%	62.0%	23.4%	14.7%	76.1%	11.4%	12.5%

**Table 3 jof-08-00383-t003:** Clinical isolates identified at the species level and reference strains included in the Filamentous Fungi Library 3.0 (*n* = 184). Number (%) of correctly identified isolates at the species level using three different sample preparation procedures. Manufacturer: when applying the manufacturer’s instructions, in-house: when applying a threshold of 1.70.

Sample Preparation Procedure	Number (%) Isolates Identified at the Species Level	
	Manufacturer	In-House
Extended direct transfer	90 (48.9%)	134 (72.8%)
Extraction	114 (62.0%)	155 (84.2%)
Liquid cultivation	140 (76.1%)	159 (86.4%)

**Table 4 jof-08-00383-t004:** Isolates identified at the genus level and included in the Filamentous Fungi Library 3.0 (*n* = 23). Results of the MALDI-TOF MS using three different sample preparation procedures. Interpretation of results, i.e., identification with high confidence (at the species level) or with low confidence (at the genus level), according to the manufacturer’s recommendations. Abbreviations: n: number, ID hc.: number of isolates identified with high confidence (at species level), ID lc.: number of isolates identified with low confidence (at the genus level), no ID: number of isolates not identified.

Clinical Isolates		MALDI-TOF MS								
		Extended Direct Transfer			Extraction			Liquid Cultivation		
Genus	*n*	ID hc.	ID lc.	no ID	ID hc.	ID lc.	no ID	ID hc.	ID lc.	no ID
*Alternaria* sp.	2	1/*A. alternata*	1	-	1/*A. alternata*	1	-	2/*A. alternata*	-	-
*Aspergillus* sp.	1	-	1	-	1/*A. nidulans*	-	-	1/*A. nidulans*	-	-
*Cladosporium* sp.	2	-	-	2	-	-	2	-	-	2
*Cunninghamella* sp.	1	-	-	1	-	-	1	-	-	1
*Fusarium* sp.	5	1/*F. proliferatum*	4	-	2/*F. proliferatum*	1	2	4/*F. proliferatum* 1/*F. equiseti*	-	-
*Lichtheimia* sp.	1	-	-	1	-	1	-	-	1	-
*Mucor* sp.	1	-	1	-	1/*M. circinelloides*	-	-	1/*M. circinelloides*	-	-
*Penicillium* sp.	3	-	1	2	-	-	3	1/*P. citrinum*	-	2
*Phaeoacremonium* sp.	1	-	-	1	-	1	-	1/*P. cinerum*	-	-
*Trichoderma* sp.	6	-	1	5	-	2	4	-	3	3
Total n	23	2	9	12	5	6	12	11	4	8
Total %	100%	8.7%	39.1%	52.2%	21.7%	26.1%	52.2%	47.8%	17.3%	34.8%

**Table 5 jof-08-00383-t005:** Isolates identified at the genus level and included in the Filamentous Fungi Library 3.0 (*n* = 23). Number (%) of isolates identified at the species level using three different sample preparation procedures. Manufacturer: when applying the manufacturer’s instructions, in-house: when applying a threshold of 1.70.

Sample Preparation Procedure	Number (%) Isolates Identified at the Species Level	
	Manufacturer	In-House
Extended direct transfer	2 (8.7%)	9 (39.1%)
Extraction	5 (21.7%)	10 (43.5%)
Liquid cultivation	11 (47.8%)	12 (52.02%)

## Data Availability

Not applicable.

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
