# Peer review of "Identification of Filamentous Fungi by MALDI-TOF Mass Spectrometry: Evaluation of Three Different Sample Preparation Methods and Validation of an In-House Species Cutoff"

_jof, 2022, doi:10.3390/jof8040383_

Round 1
Reviewer 1 Report
The manuscript evaluates and compares three different procedures for filamentous fungi identification by MALDI Biotyper. Although this has been studied before in literature, the study can be of interest for MALDI-TOF users.
The study is technically well conducted, however, it requires some revisions.
Major comments:
- Introduction, lines 62-70: some sentences described here are materials and methods of the study, and they are described again in that section. If the authors want to talk about scores, in this paragraph they should only describe the current cut-offs reccomended by the manufacturer and the cut-offs used in other studies.
- Materials and methods, section 2.4: the three methods of sample preparation does not provide enough information and they should be futher detailed. Please, describe the quantity of ethanol, formic acid and acetonitrile used, and provide details of all steps.
- I would like the authors to include some statistical analysis in order to compare the three procedures used.
- Line 116: specify which is the liquid medium used for fungi growth.
Minor comments:
- The name of microorganisms should appear in italics in Table 1 (Phaeoacremonium), Table 2A (Exophilia) and those in lines 204-206.
- Line 173: please include the complete name for A. ustus.
Author Response
Thank you for taking the time to carefully read and check the submitted manuscript and for making valuable comments/suggestions. We have addressed all raised questions (please see the responses below) and hope you will find the manuscript acceptable for publication JoF.
- Introduction, lines 62-70: some sentences described here are materials and methods of the study, and they are described again in that section. If the authors want to talk about scores, in this paragraph they should only describe the current cut-offs recommended by the manufacturer and the cut-offs used in other studies.
Response (R): We agree that the description of cutoffs and their interpretation belongs to the material and methods and is therefore redundant. According to your recommendation we deleted lines 62 -70.
- Materials and methods, section 2.4: the three methods of sample preparation does not provide enough information and they should be futher detailed. Please, describe the quantity of ethanol, formic acid and acetonitrile used, and provide details of all steps.
R: Thank you for the important advice! Each of the three mentioned sample preparation methods have now been described in more detail and can be found in lines 123-144.
- I would like the authors to include some statistical analysis in order to compare the three procedures used.
- The percentage of the identification rates of the total number of isolates had already been given in tables 2A, 2B, 3A and 3B. As the number of species is too low to perform a impactful statistical analysis this analysis would not give a more powerful insight. Therefore, we would prefer to omit additional statistical analysis.
- Line 116: specify which is the liquid medium used for fungi growth.
- The liquid medium used was Sabouraud medium. This is now stated in line 139.
Minor comments:
- The name of microorganisms should appear in italics in Table 1 (Phaeoacremonium), Table 2A (Exophilia) and those in lines 204-206.
R: This has been adapted as recommended.
- Line 173: please include the complete name for A. ustus.
R: As required we replaced of A. ustus with Aspergillus section Usti.

Reviewer 2 Report
The manuscript is well written and scientifically sound. The topic is of relevance for clinical microbiology and mycology. It is helpful for readers to learn more about the performance of the method described in such a detail. Especially the strengths and weaknesses of the different preparation methods of the MALDI TOF workflow are well documented. The manuscript is well structured, and tables are adequate to present the data generated by the authors.
The only criticism which may lead to an improvement is that the term identification at species and genus level which is first mentioned in line 63 and in the subsequent text should not be used for the MALDI TOF identification. It is adequate for the macroscopic and microscopic method which was used as gold standard in this study. However, the manufacturer of the MALDI TOF instrument, Bruker Daltonics does not name the different levels of identification anymore like this. The terms used in the User Manual are now low confidence and high confidence identification. This is more accurate because similarities in Main Spectra (MSP) do not always reflect taxonomical hierarchy, which are anyway undergoing changes and are usually based on polyphasic approaches. Therefore, my recommendation is to exchange the term genus level identification by low confidence identification and species level identification by high confidence identification.
Author Response
Thank you for taking the time to carefully read and check the submitted manuscript and for making valuable comments/suggestions. We have addressed all raised questions (please see the responses below) and hope you will find the manuscript acceptable for publication JoF.
Thank you very much for the very important advice. Accordingly we changed the wording as recommended.
